# Social Determinants of the Transition in Food Consumption in Paraíba, Brazil, Between 2008 and 2018

**DOI:** 10.3390/nu17152550

**Published:** 2025-08-04

**Authors:** Sara Ferreira de Oliveira, Rodrigo Pinheiro de Toledo Vianna, Poliana de Araújo Palmeira, Flávia Emília Leite de Lima Ferreira, Patrícia Vasconcelos Leitão Moreira, Adélia da Costa Pereira de Arruda Neta, Nadjeanny Ingrid Galdino Gomes, Eufrásio de Andrade Lima Neto, Rafaela Lira Formiga Cavalcanti de Lima

**Affiliations:** 1Postgraduate Program in Nutritional Sciences, Interdisciplinary Center for Studies in Health and Nutrition, Federal University of Paraíba, João Pessoa 58051-900, PB, Brazil; palmeirapoliana@gmail.com; 2Graduate Program in Decision Models and Health, Interdisciplinary Center for Studies in Health and Nutrition, Federal University of Paraíba, João Pessoa 58051-900, PB, Brazil; vianna@ccs.ufpb.br (R.P.d.T.V.); nadjeanny_ingrid@hotmail.com (N.I.G.G.); 3Postgraduate Program in Public Health, Interdisciplinary Center for Studies in Health and Nutrition, Federal University of Paraíba, João Pessoa 58051-900, PB, Brazil; flaemilia@gmail.com; 4Department of Nutrition, Interdisciplinary Center for Studies in Health and Nutrition, Federal University of Paraíba, João Pessoa 58051-900, PB, Brazil; patriciamoreira1111@hotmail.com; 5Foundation Building, University of Liverpool, Brownlow Hill, Liverpool L69 7ZW, UK; adeliacpereira@gmail.com; 6Graduate Program in Decision Models and Health, Federal University of Paraíba, João Pessoa 58051-900, PB, Brazil; 7Postgraduate Program in Nutritional Sciences, Graduate Program in Decision Models and Health, Interdisciplinary Center for Studies in Health and Nutrition, Federal University of Paraíba, João Pessoa 58051-900, PB, Brazil; rafaelanutri@gmail.com

**Keywords:** food consumption, eating behaviour, basic food, nutritional surveys, socioeconomic factors, health inequities, social inequalities

## Abstract

Background/Objectives: Dietary patterns have changed over time, characterising a process of nutritional transition that reflects socioeconomic and demographic inequalities among different populations. This study assessed changes in dietary consumption patterns and the associated social determinants, comparing two time periods in a sample of individuals from a state in the Northeast Region of Brazil. Methods: Data from the 2008–2009 and 2017–2018 Household Budget Survey for the state of Paraíba were analysed, totalling 951 and 1456 individuals, respectively. Foods were categorised according to the NOVA classification and compared based on sociodemographic and economic variables. To determine the factors that most strongly explain the contribution of each NOVA food group to the diet, beta regression analysis was conducted. Results: Differences were observed between the two periods regarding the dietary contribution of the NOVA food groups, with a decrease in consumption of unprocessed foods and an increase in ultra-processed foods. Living in urban areas, being an adolescent, and having an income above the minimum wage were associated with reduced intake of unprocessed foods in both periods. Additionally, being an adolescent and having more than eight years of schooling were associated with higher consumption of ultra-processed foods. Conclusions: The population under study showed changes in food consumption, reflecting a transition process that is occurring unevenly across socioeconomic and demographic groups, thereby reinforcing social inequalities. These findings can guide priorities in food and nutrition policies, highlighting the need for intervention studies to evaluate the effectiveness of such actions.

## 1. Introduction

The dietary intake of different population groups has been evaluated over the years, and changes in eating habits have become increasingly evident. These alterations reflect the process of nutritional transition, characterised by shifts in dietary patterns and nutritional status within a population, occurring in distinct ways across countries, primarily due to differing socioeconomic conditions [1,2].

In developing countries such as Brazil, the process of nutritional transition typically begins among higher-income groups. However, as economic development progresses, this pattern changes and increasingly affects the most vulnerable segments of the population, who consequently gain greater access to unhealthy foods [3]. The social distribution of the various forms of malnutrition is, in turn, influenced by determinants such as access to financial resources, information, culture, advertising, and the quality of available foods [4].

The consumption of ultra-processed foods has become increasingly prevalent worldwide [5], as demonstrated by studies investigating the relationship between food consumption and socioeconomic and demographic factors. Research involving nationally representative samples from high-income countries indicates that this food group accounts for more than half of total energy intake [6], whereas in middle-income countries, this contribution may reach up to 30% [7]. In Brazil, the rise in the consumption of ultra-processed foods has been more pronounced among individuals with higher incomes and those residing in urban areas and metropolitan regions in the south and southeast of the country [8].

Aiming to promote healthier eating habits, the Brazilian government developed the Dietary Guidelines for the Brazilian Population, introducing a new classification of foods based on their level of processing. According to this classification, the guidelines recommend that diets be primarily composed of unprocessed or minimally processed foods [9,10].

Understanding changes in dietary consumption is fundamental to guiding public food and nutrition policy planning and implementation. However, most available studies are based on national samples [8,11,12], which may not adequately reflect this process in states with specific income, education, cultural, and health characteristics, such as Paraíba. According to the most recent census, the population of Paraíba accounts for approximately 2% of the Brazilian population [13].

It is suggested that Paraíba is experiencing a slower change in food consumption compared to other states and regions of the country, considering that climatic, economic, and cultural factors vary across Brazil and influence the eating habits of its population. Moreover, the Northeast Region—one of the poorest in Brazil—has one of the lowest average intakes of ultra-processed foods [11].

Therefore, the present study aims to compare the dietary intake of individuals living in a state in the northeast of Brazil, based on the NOVA classification, between 2008 and 2018, and to analyse the associated socioeconomic and demographic determinants.

## 2. Materials and Methods

### 2.1. Study Population

The data analysed in this study were drawn from the two most recent editions of the *Household Budget Survey* (Pesquisa de Orçamentos Familiares—POF), conducted in 2008–2009 and 2017–2018 by the Brazilian Institute of Geography and Statistics (IBGE). Both editions are subsamples of a shared IBGE survey registry, which comprises several census tracts selected from all federal units across Brazil. In both survey periods, a two-stage cluster sampling design was adopted. In the first stage, census tracts were randomly selected from the shared survey registry, also referred to as the master sample. In the second stage, for each tract selected in the first, households in urban and rural areas were also randomly chosen [14].

This study focused on a subsample from the state of Paraíba. Situated in the eastern part of Brazil’s Northeast Region, Paraíba comprises 223 municipalities and covers a territorial area of 56,469.74 km^2^. In terms of the Human Development Index (HDI), Paraíba ranked 21st among the 27 federal units in 2021, with an HDI of 0.698. Approximately 68.16% of the state’s municipalities had an HDI between 0.500 and 0.599, which is considered low [15,16].

The initial sample consisted of 959 individuals surveyed in 2008–2009 and 1470 individuals surveyed in 2017–2018. However, due to missing values in socioeconomic and demographic variables, as well as the low number of individuals who self-identified as Asian or indigenous, or who did not declare any race or ethnicity, these participants were excluded from the final sample. As a result, the analytical sample included 951 individuals in the 2008–2009 edition and 1456 in the 2017–2018 edition (Appendix A).

The data used in this study are secondary and publicly available through official Brazilian government data platforms. It is important to note that the primary data collection for the Household Budget Survey (POF) was conducted by IBGE in accordance with national research standards and ethical guidelines. The survey followed a rigorously defined methodology, as documented in official technical manuals and reports [17,18].

### 2.2. Data Collection

The POF surveys were conducted over a one-year period. During this time, data were collected through interviews carried out in private permanent households, based on seven standardised questionnaires, over the course of nine days. At the beginning of each week, a field agent interviewed residents of newly selected households. Following this schedule, all households in the sample were visited over the twelve-month period [18].

The data collected through the questionnaires were organised and grouped into folders referred to as records. For this study, two records were used: the Resident record, which contains socioeconomic and demographic data collected via Questionnaire 1; and the Consumption record, which includes information on the dietary intake of each resident aged 10 years or older, obtained through Questionnaire 7 [18].

### 2.3. Dietary Intake

Dietary intake data were obtained from two food records in 2008–2009 and two non-consecutive 24 h dietary recalls in 2017–2018. In both survey years, individuals reported all foods and beverages consumed, along with detailed information regarding the characteristics of the reported items, including the preparation method, quantity, place, and time of consumption [19,20].

When analysing the impact of differences in dietary data collection instruments between the two periods, one study [21] found that the methodological changes had minimal influence on the mean estimates of energy and macronutrient intake. In the case of micronutrients, only vitamins showed significant variations. Thus, despite the shift in data collection methods—from food records in 2008–2009 to 24 h recalls in 2017–2018—studies support the viability of producing consistent estimates and valid comparisons between the two national surveys [1,11,22]. Therefore, such methodological differences do not compromise the classification of consumed foods into groups, which is the approach adopted for dietary intake analysis in the present study.

The reported foods in both POF editions were converted into energy and nutrient values using the Brazilian Food Composition Table (TBCA), version 7.0, developed by the Brazilian Network on Food Composition Data at the University of São Paulo and the Food Research Centre [23]. All foods were classified according to the NOVA classification [10], which categorises them into four groups based on their level of processing: (1) unprocessed or minimally processed foods; (2) culinary ingredients; (3) processed foods; and (4) ultra-processed foods.

To identify the individual ingredients present in preparation methods such as sautés, stews, and breaded dishes, as well as in culinary preparations like cakes, soups, beans, rice, and pasta dishes, it was necessary to disaggregate the items in order to enable their proper classification according to the NOVA system [24,25].

To better characterise the foods within the NOVA categories, the consumed items were also grouped into NOVA subgroups. In the 2008–2009 dataset, foods were classified into 58 subgroups, whereas in the 2017–2018 edition, 62 subgroups were formed (Appendix A).

Usual intake was estimated using the Multiple Source Method (MSM), a statistical software tool designed to estimate individual usual intake from repeated dietary measurements over a defined period [26]. Food groups were described according to their percentage contribution to total dietary energy intake, calculated using the following formula: (kcal from food group × 100)/total energy intake (kcal). For the analysis of subgroup consumption, the energy contribution was also calculated but relative to the total energy intake within each NOVA group.

### 2.4. Socioeconomic and Demographic Variables

The selection of socioeconomic and demographic variables from the Resident record was guided by their relevance for addressing the study objectives and their availability in both survey editions. The variables included in the analysis were household location (rural or urban), sex assigned at birth (male or female), age group (adolescents: 10–19 years; adults: 20–59 years; older adults: 60 years or older), race/skin colour (White, Brown/Black), years of schooling (more than 8 years; 1 to 7 years; or none), and per capita household income (up to ¼ of the minimum wage; between ¼ and ½; between ½ and 1; or more than 1 minimum wage). Per capita income is calculated by dividing the total household income—obtained by summing the gross monetary earnings of all household members—by the number of residents.

### 2.5. Data Analysis

The microdata were processed using the R programming language in the RStudio interface (version 4.4.1). For data manipulation, analysis, description, and visualisation, the following packages were implemented in the script: readxl, dplyr, gtsummary, flextable, ggplot2, and betareg. The comparison of the proportions of socioeconomic and demographic variable frequencies between the two survey editions was performed using the chi-square test. Possible differences between the 2008–2009 and 2017–2018 editions of the POF, considering sociodemographic variables and the mean energy contribution of NOVA groups, were analysed using independent samples *t*-tests.

To assess changes in the mean energy contribution of food groups between the two periods, absolute delta and relative delta values were calculated. These were defined as the difference between the means in 2008–2009 and 2017–2018 and the ratio of that difference to the 2008–2009 mean multiplied by 100, respectively. To identify the sociodemographic variables that best explained the contribution of each NOVA food group to the diet, univariate beta regression was used to select variables to be included in the multiple beta regression model. The models were evaluated using residual analysis and were also checked for compliance with the model’s assumptions and prerequisites. Regressions were performed separately for each year of the study. For all statistical analyses, a significance level (alpha) of 5% was adopted.

According to Resolution No. 466 of 12 December 2012 from the National Research Ethics Commission (CONEP), studies using publicly available secondary data that do not identify individual participants—such as the present study—do not require approval from a local Research Ethics Committee within the CEP/CONEP system [27].

## 3. Results

The total sample, considering both periods of study, consisted of 2407 individuals. The majority were urban residents (78%), adults (61%), and female (53%), and identified as Brown/Black (65%). Most had 1 to 7 years of schooling (44%), and 30% reported a household income above one minimum wage.

The findings of this study indicate an increase in the population’s mean caloric intake, rising from 1624 kcal in 2008–2009 to 1761 kcal in 2017–2018. When comparing the energy contribution of each food group across the two periods, statistically significant differences (*p* < 0.05) were observed in all groups within the NOVA classification. Notably, there was a reduction in the consumption of unprocessed foods and an increase in the intake of ultra-processed foods (Table 1).

In 2008–2009, the four food subgroups that contributed the most to total daily caloric intake were rice, beef, French rolls, and beans. In the second survey, the leading contributors were rice, French rolls, sugars, and poultry. When comparing the top 20 subgroups contributing to total energy intake in both editions, most were repeated, with the only differences in ranking based on the total number of calories consumed. However, in 2017–2018, two ultra-processed foods (margarine and processed meats/sausages) and one minimally processed item (pork) entered the list of the top 20 subgroups. Meanwhile, vegetables, butter, and fish were no longer among the top 20 contributors in the 2017–2018 edition (Figure 1).

In both survey editions, the number of urban residents exceeded that of rural residents, with a frequency of 73% in 2008–2009 and 82% in 2017–2018. Regarding age distribution, more than half of the participants in both editions were between 20 and 59 years of age. In terms of educational level, in 2008–2009, 53% of participants had between 1 and 7 years of schooling, whereas in 2017–2018, 50% had more than 8 years. Approximately 15% of households in 2008–2009 had a per capita income of up to ¼ of the minimum wage, while in 2017–2018 this proportion increased to 21% (Figure 2).

When comparing the mean contribution of unprocessed foods to total energy intake across both periods, stratified by each category of the socioeconomic and demographic variables, statistically significant differences (*p* < 0.05) were found in all categories. A consistent reduction in the contribution of unprocessed foods was observed in the second period compared to the first. The greatest reduction (Δ = –10%) was identified among individuals with a per capita household income of up to ¼ of the minimum wage (Table 2).

In the case of culinary ingredients, their contribution to dietary energy increased across all socioeconomic and demographic categories. This increase ranged from 4.6% among individuals with no formal education to 6.9% among women (Table 2).

The mean contribution of processed foods showed a significant reduction between the two periods across most variables. No significant changes were observed among older adults, Brown/Black individuals or those with no formal education. In both rural and urban areas, the reduction was 1.6%. When comparing across age groups, the largest decrease was observed among adolescents (3.3%). Only women showed a significant change in caloric contribution over time. Similarly, when examining race/skin colour, a significant difference was observed only among those who self-identified as White. Regarding years of schooling, only the group with no formal education did not show a difference between the two years. Finally, in terms of per capita household income, two groups showed significant differences between the years: those earning between ¼ and ½ of the minimum wage, and those earning more than one minimum wage. No differences were found among individuals with income of up to ¼ of the minimum wage or between ½ and one minimum wage (Table 2).

As for ultra-processed foods, a comparison of the two periods showed that the mean dietary contribution of these products increased across nearly all socioeconomic categories in which statistically significant differences were observed. The only groups that did not show significant changes over time were female participants; those with more than eight years of education; and those with a per capita household income above one minimum wage (Table 2).

In 2008–2009, when analysing the explanatory effect of the categories of sociodemographic variables, statistically significant differences (*p* < 0.05) were found across the variables household location, age, sex, race/skin colour, years of schooling, and per capita income, in relation to the energy contribution of unprocessed foods (Table 2). Among the categories associated with the consumption of these foods, those that best explained their caloric contribution to the diet were urban residence, adolescents, male residents, more than eight years of schooling, per capita income up to ¼ of the minimum wage, and per capita income above one minimum wage. Among these, urban residence, more than eight years of schooling, and per capita income above one minimum wage were associated with a reduction in the consumption of unprocessed foods (Table 3).

Processed food consumption in 2008–2009 was significantly explained by several sociodemographic variables. Statistically significant differences were found for household location, race/skin colour, years of schooling, and per capita income (Table 2). The categories with the strongest explanatory effect were urban residence, no formal education, per capita income up to ¼ of the minimum wage, and per capita income above one minimum wage (Table 3).

Regarding ultra-processed foods in the same period, consumption was significantly associated with household location, age, sex, and years of schooling (*p* < 0.05) (Table 2). However, only three categories had a meaningful explanatory effect: adolescents, male sex, and more than eight years of schooling. Being an adolescent or having more than eight years of schooling was associated with an increase in consumption, while being male was associated with a decrease (Table 3).

In 2017–2018, the mean caloric contribution of unprocessed foods varied significantly across almost all sociodemographic categories, except race/skin colour (Table 2). During this period, the categories that most influenced consumption were urban residence, adolescents, older adults, male sex, no formal education, per capita income between ½ and 1 minimum wage, and per capita income above one minimum wage. Among these, urban residence, adolescence, and higher per capita income were negatively associated with consumption of unprocessed and minimally processed foods. Conversely, being an older adult and having no formal education were linked to higher consumption of these foods (Table 3).

For culinary ingredients in 2017–2018, the variables household location, sex, and race/skin colour showed statistically significant differences, with the categories rural residents, female sex, and White self-identified individuals presenting the highest mean contributions (Table 2). The categories urban residence, male sex, and Brown/Black self-identified individuals had a greater explanatory effect, which was negative, indicating a reduction in consumption in this group (Table 3).

Regarding processed foods during this period, two of the six variables under study showed statistically significant differences (*p* < 0.05) in mean values across their categories: household location, with a higher mean contribution among urban residents, and per capita income, with a higher mean among those earning more than one minimum wage (Table 2). Only two categories demonstrated explanatory power in relation to consumption: urban residence, with a positive effect, indicating an increase in consumption; and per capita income up to ¼ of the minimum wage, with a negative effect, indicating a decrease in consumption (Table 3).

In the case of ultra-processed foods in 2017–2018, the variables that showed statistically significant differences in mean values, in relation to their explanatory effect on consumption, were household location, age, years of schooling, and per capita income (Table 2). Five sociodemographic categories were most strongly associated with consumption in this group: urban residence, adolescents, older adults, more than eight years of schooling, and per capita income between ½ and 1 minimum wage. Among these, only the older adult category was associated with a reduction in consumption, while all other categories were associated with an increase in the intake of ultra-processed foods (Table 3).

## 4. Discussion

The results of this study revealed an increase in daily caloric intake among the population of Paraíba between 2008 and 2018, alongside a reduction in the proportion of unprocessed and minimally processed food consumption, and an increase in the proportion of ultra-processed food consumption. These findings confirm a trend observed both nationally and internationally: the growing contribution of ultra-processed foods to family diets [5,22,28].

Although a decline in the contribution of unprocessed foods was observed, consumption of these items in Paraíba remains slightly above the national average (54.4% vs 53.2%), while the contribution of ultra-processed foods remains proportionally lower than the national average in 2017–2018 (16.3% vs 19.7%) [11]. These variations suggest that the dynamics of nutritional transition differ across regions of Brazil and, more specifically, across states. This likely results from differences in the elements that make up food systems, including industrialisation, globalisation, commercial practices, food culture, and food policies, which may give rise to region-specific patterns of purchasing behaviour and, consequently, food consumption [5,28].

The consumption of ultra-processed foods is associated with higher overall caloric intake and poorer nutritional quality, characterised by increased levels of added sugars and saturated fats and reduced intake of fibre and essential nutrients [29]. Moreover, the consumption of these products has been linked to a higher risk of chronic diseases, obesity, and premature mortality. A study analysing national dietary intake and mortality data from eight countries found that the proportion of total energy intake attributed to ultra-processed foods ranged from 15% to approximately 55%. In this context, premature mortality attributable to the consumption of ultra-processed foods ranged from 4% to around 14% of deaths. In other words, even relatively low levels of intake have a significant impact on population health. Furthermore, the higher the consumption, the greater the effect on mortality. It is estimated that, in Brazil, 57,000 premature deaths in 2019 were attributable to the consumption of ultra-processed foods, which makes the observed upward trend in their intake particularly concerning [30,31,32].

When comparing the two periods, it is possible to observe that, in the first period, individuals with higher income and education levels obtained a higher caloric contribution from ultra-processed foods compared to other groups. In the second period, however, the caloric contribution of these products increased across all income levels, with the most notable rise occurring among individuals with lower income or lower levels of education.

The rise in the contribution of ultra-processed foods may be partly explained by the increase in the proportion of urban residents observed between the two periods as these products tend to be more available and accessible in urban areas [5]. It may also reflect reduced time available for meal preparation in such settings [33], as well as the growth in the number of households classified within lower income brackets, which may have limited access to more expensive foods, leading to a shift towards lower-cost alternatives, often represented by ultra-processed products [34,35].

This trend may be attributed to the relative reduction in the prices of ultra-processed foods, which may have increased access to these items among more vulnerable populations. Fresh or minimally processed foods tend to be more expensive than ultra-processed alternatives. In Brazil, the prices of healthier foods have risen above the average for all food groups and significantly higher than those of ultra-processed products [36]. In 2008, families with higher economic status paid more for their purchases, whereas households in the Northeast Region—where Paraíba is located—and in rural areas paid lower prices [34]. In Paraíba, the dietary pattern of main meals is traditionally based on unprocessed or minimally processed foods, such as beans, rice, roots, and tubers.

The decline in the caloric contribution from unprocessed foods mirrored the increase in ultra-processed food consumption, with the largest reductions occurring among groups with lower income or education levels. This pattern is consistent with the nutritional transition phenomenon described in the literature, which involves the gradual replacement of unprocessed foods by ultra-processed products [8,22]. It also supports the idea that the increase in consumption of such products initially began among higher socioeconomic groups. Similar dynamics have been observed with other changes in dietary behaviours—for instance, the replacement of breastfeeding with formula feeding, which first became common among wealthier women and was later adopted by the rest of the population, ultimately impacting the poorest groups most, who also make up the largest share of the population [37].

These shifts in dietary patterns, whether the increase in the consumption of ultra-processed foods or of commercial infant formulas, which are also classified as ultra-processed, are strongly driven by marketing strategies employed by large corporations. This highlights the urgent need for market regulation and restrictions on advertising to preserve the population’s right to healthy eating [38]. Health-focused advertising strategies, such as labels claiming “Source of vitamins” or “Rich in minerals,” combined with the convenience and accessibility of such products may gradually influence dietary choices, leading to the normalisation of highly processed food consumption [39].

Processed foods showed smaller variations in consumption over the period studied, with only slight reductions observed in a few population groups. Although these foods are considered less harmful than ultra-processed products and are recommended for moderate consumption according to the Brazilian Dietary Guidelines, the distinction between processed and ultra-processed foods is not always clear to consumers. As a result, processed foods may be unintentionally promoted through marketing campaigns targeted at ultra-processed products, while simultaneously being discouraged by official dietary guidance [9]. These issues warrant further investigation in future studies.

The noticeable increase in the contribution of culinary ingredients observed in this study may be largely explained by the consumption of added sugars, which did not appear among the top ten energy-contributing foods in 2008, but rose to become the third highest in 2018. However, this increase should be interpreted with caution as methodological differences in measuring sugar consumption were present between the survey years [21].

The four multiple beta regression models for each food group, according to the NOVA classification, reveal independent effects of the social determinants analysed. Place of residence and years of schooling influenced all food groups. Sex and income affected three out of the four groups—processed foods and culinary ingredients were not impacted by these respective determinants. Age and race/skin colour had a lower impact on the variations in food consumption across the analysed groups. Education level, per capita household income, place of residence, and sex influence the extent of these dietary shifts, acting as social determinants of the nutritional transition and, consequently, of population health. These findings confirm the complex network of social determinants that shape food consumption patterns [40].

The study’s main strengths include its detailed description of the nutritional transition process and its demonstration of how this process occurs unequally across different socioeconomic and demographic groups. The NOVA classification is a key element in this study as it is a widely recognised benchmark for evaluating the quality of diets and their effect on different types of malnutrition. According to a systematic review [41], this classification was employed in 95% of studies examining the relationship between diet and health published between 2015 and 2019.

In addition to its scientific relevance, the NOVA classification has also been incorporated into public health policy development. Several Latin American countries, for instance, have formulated dietary guidelines based on this classification. Similarly, the French government adopted NOVA as the basis for its target of reducing ultra-processed food consumption by 20% [42,43,44]. The integration of NOVA into public health policies and programmes highlights its practical importance and potential to promote healthier eating habits.

The analysis of data from Paraíba allows for the identification of local realities and specific relationships that deepen the understanding of the issue—insights that would not be possible through the analysis of aggregated national data, given that Paraíba represents only a small portion of the country’s total population [13]. On average, Paraíba still presents a more favourable scenario compared to national figures [28]. However, further studies are needed to assess whether these differences will disappear, leading this less advanced, and currently healthier, stage of nutritional transition to converge with the dietary patterns seen in more developed regions of Brazil.

Understanding the specific situation of each location helps to plan and implement more assertive and effective interventions, particularly through the decentralization of actions to the state and municipal levels.

## 5. Limitations

The main limitation of this study lies in the comparison of dietary intake across two periods that involved certain methodological differences in the secondary data sources analysed—such as changes in the food composition tables, modifications in food recording methods, and the measurement of fats and sugars. However, the use of these datasets has been validated by national studies [1,11,22,45], which ensures both the reliability of the analysed data, and the comparisons made between the two periods.

Furthermore, the study is based on two cross-sectional surveys, which means that causal relationships cannot be confirmed, and only associations between the variables can be observed. As the populations surveyed in each period differ, it is also not possible to assess the effects of generational cohorts, which are known to influence dietary changes and directly impact the nutritional transition [46].

A key strength, however, is the use of data representative of the population, which enables both the description and comparison of the various groups analysed. The focus on residents of Paraíba allows for a deeper understanding of regional specificities that are often obscured in nationally aggregated analyses. Such insights can also contribute to understanding patterns observed in other regions and states of Brazil.

## 6. Conclusions

This study described the dietary intake of the population in the state of Paraíba at two distinct points in time, identifying changes such as a decrease in the caloric contribution from unprocessed and minimally processed foods and a simultaneous increase in the consumption of ultra-processed foods. These trends were differently influenced by factors such as education level, sex, household per capita income, and place of residence—key social determinants of the nutritional transition and, consequently, of population health.

The findings suggest that, compared to Brazil as a whole, the nutritional transition in Paraíba is at a less advanced stage. This represents an opportunity to develop targeted control measures and nutrition education strategies, as well as regulatory actions related to the food market and advertising, aimed at reducing the risks of non-communicable diseases associated with poor dietary habits. Further intervention studies are recommended to evaluate the effectiveness of such measures in improving the population’s diet.

## Figures and Tables

**Figure 1 nutrients-17-02550-f001:**
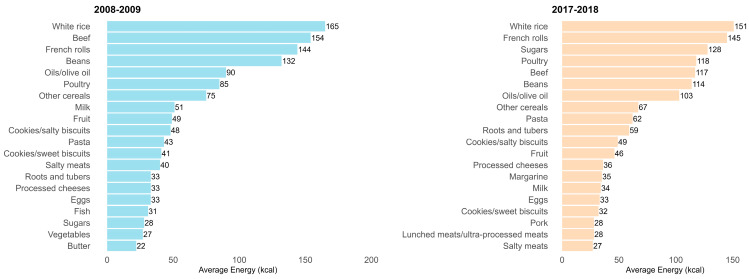
Contribution of food subgroups to total energy intake among individuals aged 10 years or older. Data from the state of Paraíba in the 2008–2009 (n = 951) and 2017–2018 (n = 1456) Household Budget Survey (POF).

**Figure 2 nutrients-17-02550-f002:**
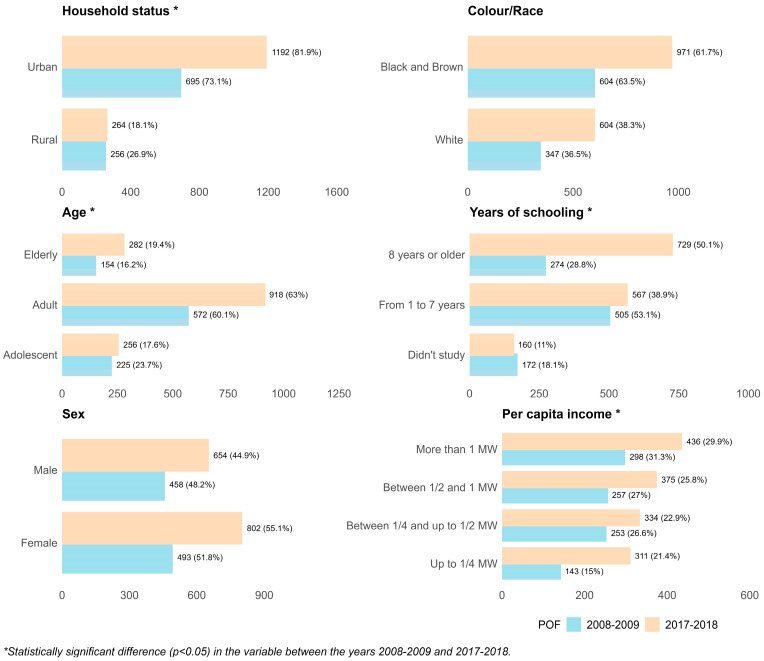
Socioeconomic and demographic variables. Data from the state of Paraíba in the 2008–2009 (n = 951) and 2017–2018 (n = 1456) Household Budget Survey (POF).

**Table 1 nutrients-17-02550-t001:** Contribution of NOVA food groups to total energy intake in the two survey editions. Data from the state of Paraíba in the 2008–2009 (n = 951) and 2017–2018 (n = 1456) Household Budget Survey (POF).

	POF Edition				
NOVA Group	2008–2009	2017–2018	Absolute Δ	Relative Δ	*p*-Value ^1^
Unprocessed	61.5	54.4	−7.1	−11.5	0.000
Culinary ingredients	9.5	15.7	+6.2	+65.2	0.000
Processed	14.8	13.6	−1.2	−8.1	0.002
Ultra-processed	14.2	16.3	+ 2.1	+14.8	0.000

^1^ The *p*-value was calculated using Student’s *t*-test for independent samples, comparing the periods 2008–2009 and 2017–2018.

**Table 2 nutrients-17-02550-t002:** Contribution of NOVA food groups to total energy intake in both survey editions, according to socioeconomic and demographic variables. Data from the state of Paraíba in the 2008–2009 (n = 951) and 2017–2018 (n = 1456) Household Budget Survey (POF).

	Share of Total Calories Consumed (%)						
	Mean						
Variables	2008–2009		2017–2018	Absolute Δ		Relative Δ	*p*-Value ^1^
	**Unprocessed**						
**Household Location ^a,b^**							
Rural	65.8		58.1	−7.7	−11.7	**0** **.000**	
Urban	59.9		53.5	−6.4	−10.7	**0** **.000**	
**Age ^a,b,^***							
Adolescent	56.9		51.2	−5.7	−10.0	**0** **.000**	
Adult	62.4		54.5	−7.9	−12.7	**0** **.000**	
Elderly	64.6		56.7	−7.9	−12.2	**0** **.000**	
**Sex ^a,b^**							
Female	60.3		53.7	−6.6	−10.9	**0** **.000**	
Male	62.7		55.2	−7.5	−12.0	**0** **.000**	
**Colour/Race ^a^**							
White	60		53.8	−6.2	−10.3	**0** **.000**	
Brown/Black	62.4		54.6	−7.8	−12.5	**0** **.000**	
**Years of education ^a,b^**							
No education	66.5		58.7	−7.8	−11.7	**0** **.000**	
1 to 7 years	62.6		55.2	−7.4	−11.8	**0** **.000**	
More than 8 years	56.3		52.8	−3.5	−6.2	**0** **.000**	
**Income per capita ^a,b^**							
Up to ¼ minimum wage	66.5		56.5	−10	−15.0	**0** **.000**	
Between ¼ and up to ½ minimum wage	62.7		55.3	−7.4	−11.8	**0** **.000**	
Between ½ and 1 minimum wage	62		52.9	−9.1	−14.7	**0** **.000**	
More than 1 minimum wage	57.6		53.3	−4.3	−7.5	**0** **.000**	
	**Processed Culinary Ingredients**						
**Household Location ^b^**							
Rural	9.9		16.5	+6.6	+66.7	**0** **.000**	
Urban	9.4		15.5	+6.1	+64.9	**0** **.000**	
**Age ***							
Adolescent	8.9		15.3	+6.4	+71.9	**0** **.000**	
Adult	9.5		15.7	+6.2	+65.3	**0** **.000**	
Elderly	10.4		16.2	+5.8	+55.8	**0** **.000**	
**Sex ^b^**							
Female	9.4		16.3	+6.9	+73.4	**0** **.000**	
Male	9.6		15	+5.4	+56.3	**0** **.000**	
**Colour/Race ^b^**							
White	9.5		16.,2	+6.7	+70.5	**0** **.000**	
Brown/Black	9.5		15.5	+6.0	+63.2	**0** **.000**	
**Years of education ^a^**							
No education	9.6		14.2	+4.6	+47.9	**0** **.000**	
1 to 7 years	9.3		15.7	+6.4	+68.8	**0** **.000**	
More than 8 years	9.4		15.9	+6.5	+69.1	**0** **.000**	
**Income per capita**							
Up to ¼ minimum wage	8.9		15.7	+6.8	+76.4	**0** **.000**	
Between ¼ and up to ½ minimum wage	9.4		15.4	+6.0	+63.8	**0** **.000**	
Between ½ and 1 minimum wage	9.5		15.8	+6.3	+66.3	**0** **.000**	
More than 1 minimum wage	10		15.9	+5.9	+59.0	**0** **.000**	
	**Processed Foods**						
**Household Location ^a,b^**							
Rural	12.1		10.5	−1.6	−13.2	**0.024**	
Urban	15.8		14.2	−1.6	−10.1	**0.001**	
**Age ***							
Adolescent	16	12.7	−3.3	−20.6	**0.000**		
Adult	14.5	13.5	−1.0	−6.9	**0.045**		
Elderly	14.2	14.5	+0.3	2.1	0.748		
**Sex**							
Female	14.9	13.5	−1.4	−9.4	**0.011**		
Male	14.7	13.6	−1.1	−7.5	0.059		
**Colour/Race ^a^**							
White	15.6	13.3	−2.3	−14.7	**0** **.000**		
Brown/Black	14.3	13.7	−0.6	−4.2	0.241		
**Years of education ^a^**							
No education	12	13.3	+1.3	10.8	0.242		
1 to 7 years	14.7	13.3	−1.4	−9.5	**0.015**		
More than 8 years	16.6	13.8	−2.8	−16.9	**0** **.000**		
**Income per capita ^a,b^**							
Up to ¼ minimum wage	11.2	11.5	+0.3	2.7	0.679		
Between ¼ and up to ½ minimum wage	15	13.4	−1.6	−10.7	**0.033**		
Between ½ and 1 minimum wage	14.3	13.8	−0.5	−3.5	0.580		
More than 1 minimum wage	16.8	14.9	−1.9	−11.3	**0.009**		
**Ultra-processed Foods**							
**Household Location ^a,b^**							
Rural	12.1	14.9		+2.8	23.1	**0** **.000**	
Urban	14.9	16.7		+1.8	12.1	**0** **.000**	
**Age ^a,b,^***							
Adolescent	18.2	20.8		+2.6	14.3	**0.007**	
Adult	13.5	16.3		+2.8	20.7	**0** **.000**	
Elderly	10.7	12.4		+2.2	20.6	**0.024**	
**Sex ^a^**							
Female	15.4	16.4		+1.0	6.5	0.052	
Male	12.9	16.2		+3.3	25.6	**0** **.000**	
**Colour/Race**							
White	14.9	16.7		+1.8	12.1	**0.013**	
Brown/Black	13.8	16.2		+2.4	17.4	**0** **.000**	
**Years of education ^a,b^**							
No education	11.1	13		+1.9	17.1	**0.023**	
1 to 7 years	13.3	15.8		+2.5	18.8	**0** **.000**	
More than 8 years	17.7	17.5		−0.2	−1.1	0.837	
**Income per capita ^b^**							
Up to ¼ minimum wage	13.4	16.2		+2.8	20.9	**0** **.000**	
Between ¼ and up to ½ minimum wage	12.9	15.9		+3.0	23.3	**0** **.000**	
Between ½ and 1 minimum wage	14.2	17.4		+3.2	22.5	**0** **.000**	
More than 1 minimum wage	15.6	15.9		+0.3	1.9	0.746	

^1^ The *p*-value was calculated using Student’s *t*-test for independent samples, comparing the periods 2008–2009 and 2017–2018. * Adolescent (10–19 years), adult (20–59 years), and older adult (60 years and over). ᵃ Statistically significant difference between categories in the univariate beta regression for the year 2008–2009.ᵇ Statistically significant difference between categories in the univariate beta regression for the year 2017–2018.

**Table 3 nutrients-17-02550-t003:** Beta estimates of explanatory variables for dietary intake according to the NOVA classification (*p* < 0.05). Data from the state of Paraíba in the 2008–2009 (n = 951) and 2017–2018 (n = 1456) Household Budget Survey (POF).

	NOVA Food Groups							
Variables	Estimates							
	Unprocessed		Culinary Ingredients		Processed Foods		Ultra-Processed Foods	
	2008–2009	2017–2018	2008–2009	2017–2018	2008–2009	2017–2018	2008–2009	2017–2018
**Household Location**								
Rural	Ref ^1^	Ref ^1^	#	Ref ^1^	Ref ^1^	Ref ^1^	Ref ^1^	Ref ^1^
Urban	−0.16888	−0.14842	#	−0.09233	+0.22965	+0.23547	*	+0.10509
**Age ^a^**								
Adolescent	−0.33406	−0.16822	#	#	#	#	+0.44045	+0.28654
Adult	Ref ^1^	Ref ^1^	#	#	#	#	Ref ^1^	Ref ^1^
Elderly	*	+0.09576	#	#	#	#	*	−0.20957
**Sex**								
Female	Ref ^1^	Ref ^1^	#	Ref ^1^	#	#	*	#
Male	+0.10242	+0.06131	#	−0.09131	#	#	−0.20947	#
**Race**								
White	Ref ^1^	#	#	Ref ^1^	Ref ^1^	#	#	#
Brown/Black	*	#	#	−0.05843	*	#	#	#
**Years of education**								
No education	*	+0.11596	+0.08905	#	−0.25280	#	*	*
1 a 7 years	Ref ^1^	Ref ^1^	Ref ^1,2^	#	Ref ^1^	#	Ref ^1^	Ref ^1^
More than 8 years	−0.22227	*	Ref ^1,2^	#	*	#	+0.32895	+0.09605
**Income per capita**								
Up to ¼ minimum wage	+0.17057	*	#	#	−0.23701	−0.15707	#	Ref ^1,2^
Between ¼ and up to ½ minimum wage	Ref ^1,2^	Ref ^1^	#	#	Ref ^1,2^	Ref ^1,2^	#	Ref ^1,2^
Between ½ and 1 minimum wage	Ref ^1,2^	−0.15126	#	#	Ref ^1,2^	Ref ^1,2^	#	+0.11094
More than 1 minimum wage	−0.15730	−0.14620	#	#	+0.12991	*	#	Ref ^1,2^

^1^ Reference variable; ^2^ Variables with *p* > 0.05, equivalent to others cited in the same category and year, considered as reference variables—not included in the multiple beta regression model; ^a^ Adolescent (10–19 years), adult (20–59 years), and older adult (60 years or older); * variables that did not explain dietary intake according to the NOVA classification (*p* > 0.05) in the beta regression model; # variables with *p* > 0.05 in the univariate beta regression and therefore not included in the multiple model.

## Data Availability

The data used in this study are publicly and freely available on the website of the Brazilian Institute of Geography and Statistics, https://www.ibge.gov.br/estatisticas/sociais/populacao/9050-pesquisa-de-orcamentos-familiares.html?edicao=9063&t=microdados (accessed on 21 May 2025).

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
