# Peer review of "Social Determinants of the Transition in Food Consumption in Paraíba, Brazil, Between 2008 and 2018"

_nutrients, 2025, doi:10.3390/nu17152550_

Round 1
Reviewer 1 Report
Comments and Suggestions for Authors
After the authors carry out the following revisions, this manuscript can be considered for publication in Nutrients journal:
Abstract: Please provide a background statement to frame your work and clearly state the main objectives; indicate some directions for further investigations at the end of the conclusions.
Include more keywords that are better related to the addressed topics of your manuscript.
The Introduction needs some improvement. A worldwide perspective of the topics to be analyzed further should be done. You also need to justify why it is important to carry out this research in Brazil. Why is your study novel and relevant?
Section 2.3 (Socioeconomic and Demographic Variables) has to be better justified and explained.
The Results are adequate. However, the quality and the size of Figures 1 and 2 can be improved.
The Discussion would benefit from a division into subsections (do the same in the Results) and a better analysis of the worldwide literature could be done. I also recommend a better discussion of the study's main limitations in a separate section.
More concrete directions for further investigations should be included at the end of the Conclusions.
Author Response
Dear Reviewer 1.
We would like to express our sincere gratitude for your insightful comments and valuable suggestions, which have significantly contributed to the improvement of our manuscript. We have implemented the recommended changes and provided our detailed, point-by-point responses to the comments below.
King regards,
Sara Ferreira de Oliveira, Rodrigo Pinheiro de Toledo Vianna, Poliana de Araújo Palmeira, Flávia Emília Leite de Lima Ferreira, Patrícia Vasconcelos Leitão Moreira, Adélia da Costa Pereira de Arruda Neta, Nadjeanny Ingrid Galdino Gomes, Eufrásio de Andrade Lima Neto, Rafaela Lira Formiga Cavalcanti de Lima.
Reviewer 1.
Comments to the Author:
“After the authors carry out the following revisions, this manuscript can be considered for publication in Nutrients journal.”
Response: We would like to thank the reviewer for their constructive feedback and for giving us the opportunity to submit our manuscript for review. We are delighted to hear that, following the requested revisions, our article may be considered for publication in Nutrients. We have carefully addressed all the points suggested for improving the manuscript.
"Abstract: Please provide a background statement to frame your work and clearly state the main objectives; indicate some directions for further investigations at the end of the conclusions."
Response:
- We have added the following contextualization to the summary: “Dietary patterns have changed over time, characterizing a process of nutritional transition that reflects socioeconomic and demographic inequalities among different populations”.
- We also reworded the main objective. This was to provide clearer information. This should facilitate understanding: “This study assessed changes in dietary consumption patterns and the associated social determinants, comparing two time periods in a sample of individuals from a state in the Northeast region of Brazil.”.
- In conclusion, we suggest future research áreas: “These findings can guide priorities in food and nutrition policies, highlighting the need for intervention studies to evaluate the effectiveness of such actions”.
“Include more keywords that are better related to the addressed topics of your manuscript.”
Response: Incluímos as seguintes palavras-chave: eating behaviour, basic food, health inequities and social inequalities.
“The Introduction needs some improvement. A worldwide perspective of the topics to be analyzed further should be done. You also need to justify why it is important to carry out this research in Brazil. Why is your study novel and relevant?”
Response: We are grateful for this constructive suggestion. We agree that the introduction would benefit from a more detailed justification. Accordingly, we have revised and expanded paragraphs 5 and 6 to provide a clearer explanation of the motivation, relevance and originality of our research. The text now provides a broader perspective on the nutritional transition in full accordance with the study's objectives. We believe this has strengthened the introduction.
“Section 2.3 (Socioeconomic and Demographic Variables) has to be better justified and explained.”
Response: To facilitate understanding, a new topic (2.2) on data collection has been added. We have also added a brief explanation of the variables at the beginning of the topic. “The selection of socioeconomic and demographic variables from the Resident record was guided by their relevance for addressing the study objectives and their availability in both survey editions.”
“The Results are adequate. However, the quality and the size of Figures 1 and 2 can be improved.”
Response: Thank you for your feedback. We have increased the resolution of the figures from 700 dpi to 900 dpi, as well as the font size of the captions.
“The Discussion would benefit from a division into subsections (do the same in the Results) and a better analysis of the worldwide literature could be done. I also recommend a better discussion of the study's main limitations in a separate section.”
Response: We appreciate the suggestion and initially considered dividing the Discussion section. We added a paragraph (lines 191–194) at the beginning of the Results section to improve readers’ understanding of the socioeconomic variables addressed and the subsequent analysis. In addition, we reorganized the Discussion section by modifying the order of the paragraphs and enhancing the clarity, detail, and alignment with the study’s main objectives. In light of these changes, we believe it is not necessary to subdivide the Results and Discussion sections into separate subsections. The study’s limitations have been presented in a separate section, as recommended.
“More concrete directions for further investigations should be included at the end of the Conclusions.”
Response: To improve the clarity of the final message, we restructured the Conclusion section. In the final paragraph, in addition to suggesting regulatory measures and public policies, we included a recommendation for future research on the topic.

Reviewer 2 Report
Comments and Suggestions for Authors
This is an interesting study but it can be improved.
- In the abstract, more and clearer results on the determinants should be presented to be in line with the title.
- It seems to us that in order to provide clearer results, multivariable regression analyses should be performed for each food group and consumption change in order to identify independent determinants.
- In the introduction, the nutrition transition should be better described.
- Why the state of Paraiba was selected should come out more clearly.
- P. 2, lines 41-45: this is unclear as a higher consumption of unhealthy diets and ultra-processed foods is forst observed in high income countries, as well as in higher income and urban residents in Brazil (see lines 71-72). Therefore, the statement that vulnerable groups are more affected is not substantiated.
- it is imperative to describe the methods used to assess dietary intake: food frequency? Recall? Over what period of time, how many days, etc? Who were the respondents?
- One wonders how the consumption of sugar (and oils and fats) as culinary ingredients was assessed, apart from what is present in foods and beverages.
- There are more not only urban, but also better educated subjects in the second survey, which may have an impact on the contribution of ultra-processed foods to energy intake.
- Table 2 needs more explanation. What is the p-value column, for instance?
- Some discussion on the NOVA classification itself is needed.
- Lines 309-311: Reference(s) needed.
- Lines 346+: The shift of a high contribution of ultra-processed foods from higher to lower income is similar to was is observed for obesity; this is an important discussion point.
Author Response
Dear Reviewer 2.
We would like to express our sincere gratitude for your insightful comments and valuable suggestions, which have significantly contributed to the improvement of our manuscript. We have implemented the recommended changes and provided our detailed, point-by-point responses to the comments below.
King regards,
Sara Ferreira de Oliveira, Rodrigo Pinheiro de Toledo Vianna, Poliana de Araújo Palmeira, Flávia Emília Leite de Lima Ferreira, Patrícia Vasconcelos Leitão Moreira, Adélia da Costa Pereira de Arruda Neta, Nadjeanny Ingrid Galdino Gomes, Eufrásio de Andrade Lima Neto, Rafaela Lira Formiga Cavalcanti de Lima.
Reviewer 2.
Comments to the Author:
“This is an interesting study but it can be improved.”
Response: We appreciate the positive comment and the careful review of our manuscript. The suggestions provided were carefully considered and incorporated whenever possible, contributing to the improvement of the study.
- “In the abstract, more and clearer results on the determinants should be presented to be in line with the title.”
Response: We summarized the results of the comparisons between the two years regarding food consumption according to the NOVA classification. We also included the results of the beta regression, highlighting the social factors that best explain the consumption of each NOVA group. Additionally, the common determinants for both years—regarding both unprocessed/minimally processed and ultra-processed foods—were synthesized in the text.
- “It seems to us that in order to provide clearer results, multivariable regression analyses should be performed for each food group and consumption change in order to identify independent determinants.”
Response: We thank the reviewer for the comment and suggestion. The multivariate regression analyses were conducted separately for each food group. We apologize for the lack of clarity in the original text. In the Discussion section, we made this information more explicit by clarifying that four distinct analyses were performed.
“The four multiple beta regression models for each food group, according to the NOVA classification, reveal independent effects of the social determinants analysed.” (Lines 417–418).
- “In the introduction, the nutrition transition should be better described.”
Response: The first paragraph of the Introduction was rewritten to provide greater clarity in the description of the nutritional transition process.
- “Why the state of Paraiba was selected should come out more clearly.”
Response: This information was described more clearly in paragraphs 5 and 6 of the introduction.
- “P. 2, lines 41-45: this is unclear as a higher consumption of unhealthy diets and ultra-processed foods is forst observed in high income countries, as well as in higher income and urban residents in Brazil (see lines 71-72). Therefore, the statement that vulnerable groups are more affected is not substantiated.”
Response: Thank you for the observation. We rewrote the second paragraph of the Introduction to avoid contradictions and took the opportunity to better clarify the concept of nutritional transition.
- “it is imperative to describe the methods used to assess dietary intake: food frequency? Recall? Over what period of time, how many days, etc? Who were the respondents?”
Response: Detailed information on the collection of dietary intake data was added to the first paragraph of section 2.3. Additionally, we included a new methodological section specifically dedicated to data collection (section 2.2).
- One wonders how the consumption of sugar (and oils and fats) as culinary ingredients was assessed, apart from what is presentin foods and beverages."
Response: We acknowledge the importance of this information. This point was clarified in the Discussion section, including a reference to a paper that has already detailed this methodology.
- “There are more not only urban, but also better educated subjects in the second survey, which may have an impact on the contribution of ultra-processed foods to energy intake.”
Response: We agree with the comment, and this aspect has been included in the Discussion section: “The rise in the contribution of ultra-processed foods may be partly explained by the increase in the proportion of urban residents observed between the two periods.”
- “Table 2 needs more explanation. What is the p-value column, for instance?”
Response: The p-value is a statistical measure that helps determine whether the results of a study are likely due to a random effect or to a real effect (e.g., a true difference or association). Moreover, the p-value quantifies the probability that the null hypothesis is true. In our case, a small p-value (p < 0.05, commonly used threshold) indicates that the observed mean difference is unlikely under the null hypothesis, providing evidence to reject it (the null hypothesis). In the opposite direction, a large p-value (p ≥ 0.05) suggests the data are compatible with no true difference.
We also added the following explanation to the footnotes of Table 1 (which also includes p-values) and Table 2: "The p-value was obtained using the Student's t-test. P-values below 0.05 indicate sufficient statistical evidence to reject the null hypothesis, suggesting a significant difference between the two survey years." In Table 2, expressions such as “p ≤ 0.05” were replaced by: “Statistically significant difference”.
- “Some discussion on the NOVA classification itself is needed.”
Response: We appreciate the suggestion to improve the Discussion section. We aimed to comprehensively address the four NOVA groups throughout the Discussion, highlighting their distinct consumption patterns and associated social determinants.
- “Lines 309-311: Reference(s) needed.”
Response: Thank you for the observation. We had not noticed the absence of the description of “Ref.” We corrected this by adding the description immediately below the table.
- “Lines 346+: The shift of a high contribution of ultra-processed foods from higher to lower income is similar to was is observed for obesity; this is an important discussion point.”
Response: We appreciate the comment, which was essential for improving the discussion on the relationship between income and the consumption of ultra-processed foods. We reorganized the text to provide a clearer explanation of this dynamic, including examples such as the replacement of breastfeeding with industrialized milk. However, we did not address the similarity with obesity, recognizing that interpreting these changes requires caution. We emphasize that this aspect deserves further investigation in future analyses, especially considering the persistent inequalities that influence food choices.

Reviewer 3 Report
Comments and Suggestions for Authors
The manuscript presents a population-based analysis of dietary changes in the state of Paraíba, Brazil. Using data from the Household Budget Surveys (POF) of 2008–2009 and 2017–2018, the study investigates the shift in caloric contribution from unprocessed to ultra-processed foods, applying the NOVA classification system. It also explores how socioeconomic and demographic variables influence this nutritional transition. The topic is highly relevant, timely, and contributes important regional evidence to the broader discussion on global dietary trends and health inequalities.
Article Report
Comments and Suggestions for Authors
Major comments:
- Introduction section: The final paragraph should explicitly state the research hypotheses or guiding questions.
- Results: Some tables (e.g., Table 2) are long and hard to interpret. Consider breaking them down by food group or summarizing the most relevant differences. Avoid repeating in the text what is already clearly shown in the tables.
- Discussion: Structure the discussion by subtopics: (1) Overall dietary trends, (2) Social inequalities, (3) Regional comparisons, (4) Policy implications. Some claims—such as the impact of advertising—should be supported with more specific and current references.
Minor comments:
- Tittle: Consider removing “Using the NOVA Classification” if it is sufficiently addressed in the abstract and body. A more concise alternative: “Social Determinants of the Dietary Transition in Brazil Between 2008 and 2018”.
- Conclusion: Briefly mention how the findings can inform targeted interventions (e.g., food education programs, regulatory policies).
- Avoid excessive repetition (e.g., “between 2008–2009 and 2017–2018” is used frequently).
- Review all tables for consistency (e.g., usage of “Ref”, “#”, “*”) and ensure all annotations are explained for the reader.

Author Response
Dear Reviewer 3.
We would like to express our sincere gratitude for your insightful comments and valuable suggestions, which have significantly contributed to the improvement of our manuscript. We have implemented the recommended changes and provided our detailed, point-by-point responses to the comments below.
King regards,
Sara Ferreira de Oliveira, Rodrigo Pinheiro de Toledo Vianna, Poliana de Araújo Palmeira, Flávia Emília Leite de Lima Ferreira, Patrícia Vasconcelos Leitão Moreira, Adélia da Costa Pereira de Arruda Neta, Nadjeanny Ingrid Galdino Gomes, Eufrásio de Andrade Lima Neto, Rafaela Lira Formiga Cavalcanti de Lima.
Reviewer 3.
Comments to the Author:
“The manuscript presents a population-based analysis of dietary changes in the state of Paraíba, Brazil. Using data from the Household Budget Surveys (POF) of 2008–2009 and 2017–2018, the study investigates the shift in caloric contribution from unprocessed to ultra-processed foods, applying the NOVA classification system. It also explores how socioeconomic and demographic variables influence this nutritional transition. The topic is highly relevant, timely, and contributes important regional evidence to the broader discussion on global dietary trends and health inequalities.”
Response: We are pleased with the positive appraisal and grateful for the careful and pertinent considerations.
Major comments:
- “Introduction section: The final paragraph should explicitly state the research hypotheses or guiding questions.”
Response: At the end of the Introduction (lines 87–89), in addition to the general objective presented in the last paragraph, we added the hypotheses that motivated the development of the study.
- “Results: Some tables (e.g., Table 2) are long and hard to interpret. Consider breaking them down by food group or summarizing the most relevant differences. Avoid repeating in the text what is already clearly shown in the tables.”
Response: We revised the description of the main columns in Table 2 and rewrote the footnotes to provide greater clarity to readers regarding the results presented. Additionally, we added a paragraph (lines 191–194) at the beginning of the Results section, describing the frequencies of socioeconomic and demographic variables. The second paragraph of this section was also rewritten to remove repetitions of information already presented in the table.
- “Discussion: Structure the discussion by subtopics: (1) Overall dietary trends, (2) Social inequalities, (3) Regional comparisons, (4) Policy implications. Some claims—such as the impact of advertising—should be supported with more specific and current references.”
Response: We appreciate the suggestion and initially considered splitting the discussion section. However, we decided to restructure it by rearranging the order of the paragraphs and improving their clarity, detail and focus, in line with the study's main objective. Given these changes, we do not believe it is necessary to subdivide the discussion into sections.
Minor comments:
- “Tittle: Consider removing “Using the NOVA Classification” if it is sufficiently addressed in the abstract and body. A more concise alternative: “Social Determinants of the Dietary Transition in Brazil Between 2008 and 2018”.”
Response: We appreciate the suggestion. We made a minor modification to the title to make it clearer that the study concerns a specific population in Brazil, as follows: “Social Determinants of the Transition in Food Consumption in Paraíba, Brazil, between 2008 and 2018”.
- “Conclusion: Briefly mention how the findings can inform targeted interventions (e.g., food education programs, regulatory policies).”
Response: We added a mention of interventions consistent with the findings, such as food education programs and regulatory policies related to the market and advertising.
- “Avoid excessive repetition (e.g., “between 2008–2009 and 2017–2018” is used frequently).”
Response: We followed the suggestion and revised the text to avoid unnecessary repetitions, particularly of that expression.
- “Review all tables for consistency (e.g., usage of “Ref”, “#”, “*”) and ensure all annotations are explained for the reader.”
Resposta: We reviewed all the tables, ensuring consistency in the use of symbols such as “Ref,” “#,” and “*,” and made sure that all annotations are properly explained in the corresponding footnotes. Accordingly, we also added the explanation for the “Ref”.

Round 2
Reviewer 2 Report
Comments and Suggestions for Authors
The authors have taken our comments into account for the revision. However, there are still a few (minor) concerns:
- The dietary methods, now described, were different in each edition (record vs recall). This should be discussed in more depth.
- In the methods section, the authors should explain how total household income was computed.
- Instead of 'race/skin color', could 'ethnic group' be considered?
- Tables 1 and 2 footnotes: It is not necessary to explain 'p' but the differences tested have to be clearly identified and with what statistic
- Is it not possible to collapse 'unprocessed' and 'minimally processed' into a single category in Table 2 for more clarity?
- Regarding 'ingredients' , it is still not clear how intake was assessed. Moreover, one wonders what is the interest of assessing determinants (Table 2)
- There is still insufficient discussion of the NOVA classification.
Author Response
Thank you for taking the time to provide feedback on the revised manuscript. We have carefully considered all of your suggestions.
Kind regards,
Sara Ferreira de Oliveira, Rodrigo Pinheiro de Toledo Vianna, Poliana de Araújo Palmeira, Flávia Emília Leite de Lima Ferreira, Patrícia Vasconcelos Leitão Moreira, Adélia da Costa Pereira de Arruda Neta, Nadjeanny Ingrid Galdino Gomes, Eufrásio de Andrade Lima Neto, Rafaela Lira Formiga Cavalcanti de Lima.
Comments to the Author:
“The dietary methods, now described, were different in each edition (record vs recall). This should be discussed in more depth.”
Response: We appreciate the suggestion and, based on it, we have included the following paragraph in the methodology section (2.3): “When analysing the impact of differences in dietary data collection instruments between the two periods, one study found that the methodological changes had minimal influence on the mean estimates of energy and macronutrient intake. In the case of micronutrients, only vitamins showed significant variations. Thus, despite the shift in data collection methods—from food records in 2008–2009 to 24-hour recalls in 2017–2018—studies support the viability of producing consistent estimates and valid comparisons between the two national surveys. Therefore, such methodological differences do not compromise the classification of consumed foods into groups, which is the approach adopted for dietary intake analysis in the present study.”
" In the methods section, the authors should explain how total household income was computed."
Response: We appreciate the suggestion. We have added this information at the end of the paragraph in Section 2.4, which now reads as follows: “Per capita income is calculated by dividing the total household income—obtained by summing the gross monetary earnings of all household members— by the number of residents.”
“Instead of 'race/skin color', could 'ethnic group' be considered?”
Response: We appreciate the suggestion and acknowledge that countries may use the terms race, ethnicity, and skin color in different ways. In the United States, for example, racial classification distinguishes between “race” and “ethnicity,” as in “Hispanic or Latino” (ethnicity) and “Black” or “White” (race). In Brazil, the population data collection question on racial identification refers to “color or race”. It includes five categories: White, Black, Brown (Pardo), Yellow (Asian), and Indigenous, aiming to reflect the country's racial diversity. In this sense, we understand that race and ethnicity are distinct concepts. For us, “Skin color” refers to phenotypic characteristics, “ethnicity” encompasses the cultural identification of human groups, and “race” is a social construct, currently understood as a sociological category marked by exclusion and racism, particularly affecting Black and Brown populations. Therefore, in Brazil, it is not possible to distinguish between Black and White individuals solely based on cultural identification. Instead, the distinction must be made from a sociological perspective, shaped by a history of racist social hierarchies. Thus, within the Brazilian context, the use of the term “race/skin color” is more appropriate.
Further information can be found in the references:
https://pmc.ncbi.nlm.nih.gov/articles/PMC9774479/
“Tables 1 and 2 footnotes: It is not necessary to explain 'p' but the differences tested have to be clearly identified and with what statistic”
Response: We also appreciate the suggestions that helped improve the tables presented. Accordingly, we have included the following footnote: “The p-value was calculated using Student’s t-test for independent samples, comparing the periods 2008–2009 and 2017–2018.”
“Is it not possible to collapse 'unprocessed' and 'minimally processed' into a single category in Table 2 for more clarity?”
Response: We appreciate the suggestion. When preparing the table, we did not realize that it could give the impression that these were distinct categories, when in fact, they are the same. To ensure greater clarity, we have revised the description of the first group analysed.
“Regarding 'ingredients', it is still not clear how intake was assessed. Moreover, one wonders what is the interest of assessing determinants (Table 2)”
Response: We appreciate the suggestion and, based on it, we have included the following paragraph in the methodology section (2.3): “To identify the individual ingredients present in preparation methods such as sautés, stews, and breaded dishes, as well as in culinary preparations like cakes, soups, beans, rice, and pasta dishes, it was necessary to disaggregate the items in order to enable their proper classification according to the NOVA system.”
We understand that the data in Table 2 are crucial for a better understanding of, and discussion about, the nutritional transition process between social and demographic groups.
“There is still insufficient discussion of the NOVA classification.”
Response: We appreciate the comment and acknowledge the importance of providing a more in-depth discussion of the NOVA classification. Therefore, we have added two paragraphs (12 and 13) at the end of the discussion section.
